# Spatial Immunology in Liver Metastases from Colorectal Carcinoma according to the Histologic Growth Pattern

**DOI:** 10.3390/cancers14030689

**Published:** 2022-01-29

**Authors:** Gemma Garcia-Vicién, Artur Mezheyeuski, Patrick Micke, Núria Ruiz, José Carlos Ruffinelli, Kristel Mils, María Bañuls, Natàlia Molina, Ferran Losa, Laura Lladó, David G. Molleví

**Affiliations:** 1Tumoral and Stromal Chemoresistance Group, Oncobell Program, Institut d’Investigacions Biomèdiques de Bellvitge (IDIBELL), Gran Via 197-203, L’Hospitalet de Llobregat, 08908 Barcelona, Catalonia, Spain; ggarciav@idibell.cat (G.G.-V.); nuria.ruiz@bellvitgehospital.cat (N.R.); jruffinelli@iconcologia.net (J.C.R.); kristel.mils@bellvitgehospital.cat (K.M.); maria.banuls@iconcologia.net (M.B.); nmolina@idibell.cat (N.M.); flosa@iconcologia.net (F.L.); laurallado@bellvitgehospital.cat (L.L.); 2Department of Immunology, Genetics and Pathology, Uppsala University, S-75105 Uppsala, Sweden; patrick.micke@igp.uu.se; 3Department of Pathology, Hospital Universitari de Bellvitge, L’Hospitalet de Llobregat, 08908 Barcelona, Catalonia, Spain; 4Department of Medical Oncology, Institut Català d’Oncologia, L’Hospitalet de Llobregat, 08908 Barcelona, Catalonia, Spain; 5Department of Surgery, Hospital Universitari de Bellvitge, L’Hospitalet de Llobregat, 08908 Barcelona, Catalonia, Spain; 6Program Against Cancer Therapeutic Resistance (ProCURE), Institut Català d’Oncologia, L’Hospitalet de Llobregat, 08908 Barcelona, Catalonia, Spain

**Keywords:** liver metastases, lymphocyte, immunology, desmoplasia, growth pattern, multiplex

## Abstract

**Simple Summary:**

In the era of immunotherapy, the tumor microenvironment (TME) has attracted special interest. However, colorectal liver metastases (CRC-LM) present histological peculiarities that could affect the interaction of immune and tumor cells such as fibrotic encapsulation and dense intratumoral stroma. We explored the spatial distribution of lymphocytic infiltrates in CRC-LM in the context of the histologic growth patterns using multispectral digital pathology providing data on three different scenarios, tumor periphery, invasive margin, and central tumoral areas. Our results illustrate a similar poor cell density of CD8^+^ cells between different metastases subtypes in intratumoral regions. However, in encapsulated metastases, cytotoxic cells reach the tumor cells while remaining retained in stromal areas in non-encapsulating metastases. Some aspects are still unresolved, such as understanding the reason why most lymphocytes are largely retained in the capsule.

**Abstract:**

Colorectal cancer liver metastases (CRC-LM) present differential histologic growth patterns (HGP) that determine the interaction between immune and tumor cells. We explored the spatial distribution of lymphocytic infiltrates in CRC-LM in the context of the HGP using multispectral digital pathology. We did not find statistically significant differences of immune cell densities in the central regions of desmoplastic (_d_HGP) and non-desmoplastic (_nd_HGP) metastases. The spatial evaluation reported that _d_HGP-metastases displayed higher infiltration by CD8^+^ and CD20^+^ cells in peripheral regions as well as CD4^+^ and CD45RO^+^ cells in _nd_HGP-metastases. However, the reactive stroma regions at the invasive margin (IM) of _nd_HGP-metastases displayed higher density of CD4^+^, CD20^+^, and CD45RO^+^ cells. The antitumor status of the TIL infiltrates measured as CD8/CD4 reported higher values in the IM of encapsulated metastases up to 400 μm towards the tumor center (*p* < 0.05). Remarkably, the IM of _d_HGP-metastases was characterized by higher infiltration of CD8^+^ cells in the epithelial compartment parameter assessed with the ratio CD8^epithelial^/CD8^stromal^, suggesting anti-tumoral activity in the encapsulating lesions. Taking together, the amount of CD8^+^ cells is comparable in the IM of both HGP metastases types. However, in _d_HGP-metastases some cytotoxic cells reach the tumor nests while remaining retained in the stromal areas in _nd_HGP-metastases.

## 1. Introduction

Colorectal carcinoma (CRC) is the third most frequent tumor in the world, with 1,932,000 new cases annually worldwide (GLOBOCAN 2020). The incidence of this tumor increases by 2% each year and is the primary cause of death due to cancer. The number of deaths annually is approximately half that of its incidence. Therefore, tackling this disease is one of the highest priority health challenges. In addition, approximately 50% of patients will develop either synchronous or metachronous colorectal liver metastases (CRC-LM) [1], with less than 10% of 5-year survival rates in these cases [2]. Moreover, the great majority of patients with metastasis (around 70%) are treated solely with chemotherapy, biologics in a selected group of patients [3] and immunotherapy in cases of mismatch repair deficiency [4]. Therefore, it is important to improve our knowledge of these tumors particularly for the metastatic setting. 

Recent advances in immunoncology have sparked interest in the interaction of immune cells with tumor cells and other stromal cells. However, CRC-LM present histological peculiarities that could affect such interaction. CRC-LM grow according to three histological growth patterns (HGP) [5], desmoplastic (_d_HGP), pushing (_p_HGP) and replacement (_r_HGP). Pushing and replacement subtypes are normally grouped in a single class, as non-desmoplastic (_nd_HGP). The desmoplastic subtype is characterized by a dense fibrotic capsule surrounding the tumors. No contact takes place between tumor cells and hepatocytes. Around 40% of CRC-LM displayed such encapsulation in at least more than 50% of its interface with the normal parenchyma and 20% displayed a complete fibrotic rim, this group of patients being those with a very good outcome [6]. This fibrotic rim is formed by cancer-associated fibroblasts (CAFs) and a dense lympho-histiocytic infiltrate and massive matrix deposition. Using conventional histology staining, the lymphocytic infiltration in _d_HGP is easily identifiable as a well-defined belt in the most distal half of the fibrotic rim. However, its detailed characterization, as well as the biologic reason for the lack of infiltration in more central areas of the fibrotic rim and closer to the tumor has still not been fully elucidated. 

The aim of this study was to explore the spatial distribution of lymphocytic infiltrates in CRC-LM in the context of the HGPs using a multiplex immunohistochemistry (mIHC) and spatial image analysis.

## 2. Materials and Methods

### 2.1. Study Cohort/Material

Twenty-two consecutive cases of untreated CRC-LM were obtained with the approval of the Ethics Committee of the Hospital Universitari de Bellvitge (IDIBELL) after the informed consent of the patients. Thirteen cases were characterized as desmoplastic/encapsulated (_d_HGP) and nine cases as non-desmoplastic (_nd_HGP), characterized by the absence of the fibrotic rim between the liver parenchyma and the tumor. Five cases out of twenty-two were synchronous metastases, three characterized as _d_HGP and two as _nd_HGP. HGP was scored following previously reported consensus guidelines [7]. No MSI samples were included.

### 2.2. Multiplex Immunofluorescence Staining

Quantitative multiplex staining was performed in whole surgical FFPE sections.

Briefly, four µm thick sections were de-paraffinized, rehydrated and subjected to HIAR (microwave, pH9, 15 min). To enable multiplexing, the staining procedure included several iterative cycles. In each of them the primary antibody (incubation time 30 min, RT) and the secondary polymerized reporter enzyme staining system (undiluted, incubation time 10 min, RT) were applied, and followed by tyramide signal amplification and developing with Opal™ fluorophore (1:100, incubation time 10 min, RT) (Akoya Biosciences). After a target was completed, HIAR (microwave, pH6, 15 min) was applied to quench endogenous peroxidase activity, to allow both antigen retrieval and removal of antibodies and polymer system from the previous cycle. After the final staining cycle, 4′,6-diamidino-2-phenylindole (DAPI) was applied to visualize nuclei.

A detailed procedure of retrieval buffers, primary antibodies, and amplification systems are described in Appendix A.

### 2.3. Imaging

Imaging was performed with the Vectra Polaris system (Akoya Biosciences, Marlborough, MA, USA). First, each whole slide was scanned at 10× magnification. Then, the regions (1.86 × 1.39 mm) for image acquisition were selected to capture areas from the central part of the metastases and from the periphery. The selected regions were imaged in a multispectral mode at a resolution of 2 pixels per 1 μm. 

### 2.4. Image Analysis, Thresholding and Immune Cell Subclasses

Image spectral deconvolution and analysis were performed by inForm (2.4.8) software (Akoya), described in detail previously [8,9] In short, images were classified into three categories: two tissue compartments (the epithelial compartment and stroma compartment), and blank areas. DAPI nuclear staining was used to perform cell segmentation. The images were reviewed for artifacts, staining defects, and the accumulation of immune cells in necrotic areas and intraglandular structures, which were manually excluded. 

Marker intensity thresholds were defined as described before [9]. 

Because tissue-classifier did not allow liver parenchyma to be distinguished from tumor tissue, as Pan-cytokeratin stained both tumor cells and hepatocytes, manual selection on each of the regions was performed. Thus, three tissue categories were segmented: Tumor tissue (can contain both epithelial and small regions of stroma between adjacent tumor glands), liver parenchyma (can contain both epithelial and stroma compartments), and stroma, including the fibrotic capsule in _d_HGP metastases and some eventual stromal areas between liver and tumor cells in _nd_HGP metastases.

### 2.5. Spatial Analysis

We evaluated cell distribution at the tumor periphery, the invasive margin, and central areas of metastatic tissue. These demarcations were defined as follows: tumor periphery, 100 μm of the proximal liver parenchyma until the first row of tumor cells, including the capsule in _d_HGP CRC-LM; invasive margin, 800 μm from the liver-tumor interface (_nd_HGP) or capsule-tumor interface (_d_HGP) to the deeper tumor areas; central areas, deeper and distal tumor areas from the liver parenchyma. Certain tissue category(ies) (for example, liver parenchyma and stroma) were used as ‘reference tissue’, while the cells of interest were identified in other tissue categories (for example, tumor tissue). Each cell of interest was defined by marker combination as described above. For each cell of interest in the tumor tissue, the nearest neighboring cell (of any subclass) in liver parenchyma or stroma was identified and the distance was measured. To simplify data visualization and analysis, the distances were split into four zones: 0 to 100 µm, 100 to 200 µm, 200 to 400 µm, 400 to 600 µm, taking as a reference two different measures (described in Figure 1B). Thus, a certain number of the cells of interest was identified in each of the zones. To normalize the cell counts, the zone area had to be assessed. For this purpose, spatial coordinates of all cells in the zone were retrieved. Further, a convex-hull algorithm was used to generate a polygon, covering the retrieved coordinates and to calculate the polygon area. Cell count was normalized per mm [2].

### 2.6. Statistics

Statistical analyses were performed using R Studio (Version 1.3.1073). The Ward algorithm was used for hierarchical clustering and the Wilcoxon signed rank test was used to compare variables between central and peripheral of the same patient. The Mann–Whitney U test with Pratt correction for tied data was used to compare variables between independent samples. *p* < 0.05 was considered statistically significant.

## 3. Results

### 3.1. The Comparison of Immune Cell Densities between Encapsulated and Non-Encapsulated Liver Metastases

First, we aimed to compare immune infiltrating lymphocyte (TIL) densities in _d_HGP and _nd_HGP metastases. The marker combination used to characterize TIL subclasses is displayed in Figure 1A. Because of the importance of the spatial context [10], we assessed separately the central and peripheral regions of the malignant lesions (Figure 1B). No differences were observed in central areas between metastases with different HGP (Figure 1B). Surprisingly, we also did not observe substantial differences between _d_HGP and _nd_HGP metastases when analyzing total immune infiltrates (epithelial + stroma compartments) in peripheral regions (Figure 1B). When analyzing immune infiltrates in different compartments separately, we observed in _nd_HGP metastases higher infiltration of CD4^+^, CD4^+^ memory, CD45RO^+^ cells in stroma compartment (*p* = 0.041, *p* = 0.0005 and *p* = 0.032 respectively), and CD4^+^ memory cells in the epithelial compartment (*p* = 0.041).

In conclusion, we did not find statistically significant differences between immune infiltration in central regions of metastases with different HGP. In peripheral regions non-encapsulated metastases showed higher infiltration of CD4^+^ positive lymphocyte subsets and CD45R0^+^ cells.

### 3.2. Spatial Assessment of TILs in Liver Metastases

This initial analysis, however, did not take into account the differential histological characteristic represented by tumor encapsulation. Therefore, the analyzed peripheral regions in _d_HGP were mainly represented by the capsule, which retained most of the immune infiltrates, while the peripheral regions in _nd_HGP, selected for analysis, mostly consisted of tumor tissue, liver parenchyma and small areas of reactive stroma. Such a difference in histological composition makes the direct comparison of peripheral regions suboptimal and suggests establishment of more reliable methods with comparable reference elements in both HGPs. With these results, we set out to evaluate lymphocyte infiltration, considering spatial location of immune infiltrates in relation to tumor and host tissue. However, the relation between these tissue elements is not equivalent in _nd_HGP and _d_HGP. Thus, the interface between healthy and tumoral tissue is easy to identify and could serve as a reference for spatial analysis in _nd_HGP metastases. However, in metastases with _d_HGP the fibrotic rim constitutes a barrier between host liver parenchyma and tumor tissue, and it is not clear whether to consider the capsule as an element of the host or the tumor. Taking into account this histological characteristic, we considered two interface lines between different tissue types, to serve as references for spatial analysis of the immune infiltrates, allowing two measurement types: A and B, as illustrated in Figure 2. Measure A evaluates cell distance from the interface between liver parenchyma and either capsule (in _d_HGP) or tumor (in _nd_HGP) or small areas of reactive stroma (in _nd_HGP). Measure A thus includes the cells, located in the capsule in _d_HGP metastases, in reactive stroma in _nd_HGP metastases, and in tumor tissue in both HGP. Measure B excludes cells from stroma between liver parenchyma and tumor and includes only the cells from tumor tissue, thus, strictly speaking that which corresponds to the invasive margin of the tumor.

Considering Measure A, desmoplastic _d_HGP metastases displayed higher infiltration of CD8^+^, CD8^+^ single positive, and CD20^+^ lymphocytes, particularly in areas close to the liver parenchyma (Figure 3A, left panel). This distance, in encapsulated metastases, corresponds mainly to the outer capsule fragment closest to the liver, where a large quantity of TILs is retained. However, when moving away from the parenchyma towards the tumor, the differences between HGP became less prominent and at 400 μm were completely lost. These, deeper regions most likely corresponded to invasive margin in _d_HGP and deeper tumoral regions in _nd_HGP.

On the other hand, the results were different when we used Measure B. In this case, CD4^+^, CD4^+^ single positive, CD4^+^ memory, CD4_Treg (CD4^+^ and FoxP3^+^), and CD45R0^+^ TILs were significantly more abundant in non-encapsulated metastases (Figure 3A, right panel), while no difference was seen for CD8^+^ subsets.

Interestingly, the overall cell density for most of the immune subsets was lower when using Measure B not only in encapsulated metastases (which is because the majority of the immune cells in this HGP is retained in the capsule) but also in non-encapsulated metastases, probably, revealing the impact of the regions of reactive stroma, present on the tumor/liver border in non-encapsulated lesions (as an example, the infiltrates located between the white and orange lines in Figure 1B right image). This difference was especially prominent for all CD4^+^ subsets, CD20 cells, and CD45R0^+^ cells. 

Even larger prevalence of CD4^+^, CD4^+^ single positive, CD4^+^ memory, CD4_Treg, and CD45R0^+^ cells in terms of statistical significance and absolute cell counts was demonstrated in non-encapsulated metastases. Additionally, CD20^+^ B cells showed higher levels in non-encapsulated samples, thus, demonstrating cellular composition in the regions of reactive stroma.

In conclusion, the fibrotic capsule in encapsulated metastases, especially the outer capsule part is characterized by increased quantity of CD8^+^ positive lymphocyte subsets and B cells, in comparison to peripheral regions in _nd_HGP metastases. However, the invasive margin of tumor tissue in encapsulated metastases did not demonstrate enrichment of these immune cell subsets, and conversely, showed a lower density of CD4^+^ positive lymphocyte subsets and CD45R0^+^ cells. The reactive stroma regions in non-encapsulated metastases are characterized by accumulation of CD4^+^ positive lymphocyte subsets, CD45R0^+^ cells, and CD20^+^ cells.

### 3.3. The Comparison of Immune Cell Densities in the Invasive Margin of Liver Metastases

Spatial analysis revealed the importance of accurate assessment of the histological peculiarities of liver metastases with different HGP. Because we assumed that direct interaction between immune cells and malignant tissue is of major importance for immune surveillance in metastatic disease, we focused our next analyses on the metrics, derived using ‘Measure B’ approach and performed manual selection of the invasive margin regions (i.e., tumor regions, excluding capsule, liver parenchyma, and reactive stroma) in both HGP.

Non-encapsulated metastases displayed higher quantity of total CD4^+^, CD4^+^ single positive, CD4^+^ memory cells, as well as CD20^+^ B cells in the invasive margin (Figure 3B; *p* = 0.011, *p* = 0.038, *p* = 0.008 and *p* = 0.032 respectively). These differences corresponded mainly to inflammatory infiltrates circumscribed to the stromal compartment, which retained most of the identified infiltrates (Figure 3B, middle panel), with the exception of total CD4^+^ memory cells that demonstrated greater infiltration also in the epithelial compartment in non-encapsulated metastases (Figure 3B, lower panel).

In conclusion, the analysis of the invasive margin confirmed the spatial analysis and demonstrated that non-encapsulated metastases have higher infiltration of several CD4^+^ positive lymphocyte subsets.

### 3.4. The Comparison of Immune Cell Densities in the Invasive Margin and Tumor Center

Next, we compared the inflammatory infiltrates between the invasive margin and central areas of the tumor in each sample in a pairwise manner. _d_HGP displayed higher infiltration by CD4^+^ total, CD4^+^ single positive, and CD4_Treg cells in central areas compared to the invasive margin while the trend was the opposite for these cell subtypes in _nd_HGP (Appendix A). Encapsulated metastases also showed more CD20 B cells in central areas than in the invasive margin.

Thus, summarizing the results from two HGPs and two locations (central and invasive margin) we can state that the poorest immune infiltration is observed in the invasive margin of encapsulated metastases, while their central parts as well as the invasive margin and central parts of non-encapsulated metastases demonstrated comparable quantities of TILs.

### 3.5. Anti-Tumoral Status of the Immune Infiltrates in the Invasive Margin and Tumor Center

Our results did not match the established paradigm of TIL infiltration in tumors. Thus, TIL infiltration is expected to be associated with improved survival. Encapsulated metastases are also known to have substantially better prognosis than non-encapsulated. However, in our study, we observed lower TIL levels in malignant tissues of encapsulated metastases. We hypothesized, that such results may be explained by the presence of TILs with immune-suppressive functions. In fact, while CD8 lymphocytes are considered to be main tumor cell killers [11], among the CD4^+^ T helper subsets, the cells of type 2 (Th2) may dominate in tumors [12]. Th2 cells drive the polarization of macrophages towards M2-type and eventually create an immune-suppressive tumor microenvironment [13].

Thus, we sought to evaluate the overall anti-tumoral status of the TIL infiltrates, assuming that CD8^+^ cells represent anti-tumor immunity while CD4^+^ cell subsets may contain significant fraction of Th2 T cells and therefore reflect pro-tumor characteristics. To assess TIL antitumoral status we generated a metric $:\;\frac{\mathrm{CD}8}{\mathrm{CD}8+\mathrm{CD}4}$, which is denoted further in the text as ‘CD8/CD4 ratio’.

The CD8/CD4 ratio was higher in the invasive margin, than in the central areas in _d_HGP samples, while in _nd_HGP metastases the result was the opposite (Appendix A). Further, when applying spatial analysis using Measure B, i.e., excluding any peritumoral stroma, encapsulated metastases demonstrated a higher CD8/CD4 ratio at the tumor margin and up to 400 μm towards the tumor center (*p* < 0.05) (Figure 3A, right panel). At the same time, no difference was seen in the CD8/CD4 ratio between the central areas of the two HGP (Figure 4, upper panel). Finally, analysis of invasive margin regions confirmed the observations of spatial distribution and demonstrated a greater CD8/CD4 ratio in encapsulated metastases (Figure 4, upper panel). 

The active anticancer response by cytotoxic cells requires their presence in the immediate vicinity of the malignant cells, enabling their direct contact and cytotoxic function. Thus, if the CD8/CD4 ratio reflects the antitumoral immune microenvironment in _d_HGP metastases, one would expect a higher number of CD8 (cytotoxic) cells in the epithelial compartment in _d_HGP in comparison to _nd_HGP samples. Indeed, when we analyzed CD8^+^ infiltration in the invasive margin specifically in the epithelial compartment, in relation to the cytotoxic cells that remain limited in the stromal compartment, the encapsulated metastases demonstrated significantly higher levels (evaluated as $\frac{\mathrm{CD}8\;\mathrm{epithelial}}{\mathrm{CD}8\;\mathrm{epithelial}+\mathrm{CD}8\;\mathrm{stromal}};\;$Figure 4, lower panel; *p* = 0.021, Mann–Whitney U test). 

Collectively, the CD8/CD4 ratio, which we consider as a surrogate metric of the anti-tumoral status of the TIL infiltrates, was significantly elevated in the invasive margin in encapsulated metastases, in comparison to central areas and to the invasive margin in non-encapsulated lesions, potentially reflecting an anti-tumoral immune microenvironment in _d_HGP metastases. Moreover, the invasive margin in encapsulated metastases was characterized by higher prevalence of CD8^+^ cells in the epithelial compartment, suggesting the presence of an anti-tumoral microenvironment in _d_HGP lesions and the presence of an immune-suppressive, pro-tumoral microenvironment in _nd_HGP lesions.

## 4. Discussion

The present study aimed to characterize the spatial distribution of lymphocytic infiltrates in chemonaive CRC-LM in the context of the particular HGP. We analyzed the immune infiltrates in 22 untreated CRC-LM taking into account the characteristic HGP of metastases. We assessed the spatial distribution in the tissue on three different levels: (a) by distinguishing the tumor central and peripheral regions and the invasive margin, (b) by accurate measurement of immune cell distribution in relation to different histological hallmarks and (c) by the analysis of the immune infiltration in stromal and epithelial compartments. To the best of our knowledge, this is the first study, providing a systematic and reproducible approach for the topographic distribution of lymphocytic infiltrates in CRC-LM in the context of the HGP.

There are many studies related to prognostic factors in CRC-LM, whether surgical parameters [2], histopathological [14], molecular [15] or immunological parameters [16]. However, the importance of immunological parameters, such as Immunoscore [17], has recently become evident, adding to the already known predictive value of Kras mutations [18].

On the other hand, the HGP have a deep impact on the survival of patients with CRC-LM [19,20,21]. Likewise, the tumor microenvironment (TME) is of paramount relevance in the formation of such growth patterns, since in the fibrotic rim and the lymphocytic belt there are contained demarcate histologic structures that might define the immune response against the tumor. However, the retention on the outer portion of the capsule of such a large number of infiltrates and the association with a better prognosis remains to be elucidated.

In addition, it has already been published that lymphocytic infiltrates are not uniformly distributed in CRC-LM [22], and quite often immune cells tend to be located in the peripheral areas of the tumor [23,24,25]. However, in the majority of these published studies neither the HGP nor the spatial distribution was taken into consideration. More recently, Berthel A et al. [26], reported distinct patterns of immune cell infiltration in CRC-LM, revealing a certain degree of infiltrate heterogeneity depending on the distance from the invasive margin. However, no mention was made of HGP. Hoppener D et al. [27] reported that encapsulating metastases are characterized by an increase in cell densities of different cytotoxic populations, and the authors attribute this fact to the better survival of patients with these metastases. Following this observation, some authors suggested that _nd_HGP CRC-LM should be considered as *immune desert* tumors and _d_HGP as *immune excluded* [28].

The main difference in our exploratory study compared to previously reported results is that we introduced a standardized reproducible approach for spatial analysis in three different locations, (i) in central areas of the tumor, where there is usually an enrichment in a mature stroma with low tumor cellularity, (ii) in the tumor invasive margin, that is, in the outermost area with the presence of malignant cells, either in contact with hepatocytes (_nd_HGP), reactive stroma regions (n_d_HGP), or with the fibrous capsule (_d_HGP) and iii) in the peripheral area, also referred to as the outer invasive margin [29], which in CRC-LM includes the capsule (_d_HGP), reactive stroma regions (n_d_HGP), any small portions (100 μm) of peritumoral stroma, and normal liver parenchyma and small portions (100 μm) of tumor.

In our initial analysis we evaluated immune cells in peripheral areas, which were not stringently defined and observed the expected prevalence of cytotoxic cells in encapsulated metastases, thus, supporting the *immune excluded/immune desert* paradigm. However, we noticed that most of the immune cells of the peripheral regions in _d_HGP are retained in the outer part of the desmoplastic capsule and thus, never establish contacts with cancer cells, while most of the immune cells in _nd_HGP are localized in regions of reactive stroma. When we excluded these regions form the analysis, we observed higher immune cell densities in _nd_HGP lesions. Thus, we doubt that the *immune excluded/immune desert* concept is applicable for CRC-LM.

Interesting, the higher immune cell densities in the invasive margin of _nd_HGP lesions were represented by the subsets of CD4^+^ cells. Although we could not dissect the functional characteristic of these subsets, we hypothesize they may contain an increased fraction of Th2-type T helpers with immune-suppressive capacities.

This hypothesis is supported by the analysis of the CD8/CD4 ratio, which was significantly lower in non-desmoplastic tumors, and even further confirmed by the analysis of CD8^+^ cells, which were enriched in the epithelial compartment in the invasive margin in desmoplastic tumors, suggesting cytotoxic activity. Additionally, a higher CD8/CD4 ratio was measured in the central areas of _nd_HGP compared to their matched invasive margin, which could be closely related to the mechanism of CD8 siphoning described very recently [30].

Taken together, the quantity of cytotoxic CD8^+^ cells is comparable in invasive margins in both HGP, but the immune microenvironment in _nd_HGP seem to be immune-suppressive and blocks the cytotoxic activity of CD8^+^ cells. One might therefore speculate that _nd_HGP CRC-LM could benefit from immune-checkpoint therapy, which may help to re-activate pre-existing antitumoral lymphocytes. On the other hand, _d_HGP CRC-LM could be suitable candidates for immunotherapy based on adoptive T cell transfer since small amounts of cytotoxic CD8 T cells reached the tumor nests.

## 5. Conclusions

As a summary, the present study provides a description of different patterns of lymphocyte infiltration depending on the HGP. The composition and the topographic demarcation in relation to the invasive margin depicted different immune scenarios according to the HGP in CRC-LM. Perhaps the most relevant finding was the presence of immune-suppressive microenvironment in metastases with _nd_HGP, and antitumoral immune microenvironment in metastases with _d_HGP, a fact that could explain the better prognosis of encapsulating metastases. Despite this difference, both HGP are characterized by low levels of cytotoxic lymphocytes and thus, could benefit from immune-therapies. Some aspects remain unresolved for now, such as understanding the reason for the retention of a large number of lymphocytic infiltrates in the external part of the capsule.

## Figures and Tables

**Figure 1 cancers-14-00689-f001:**
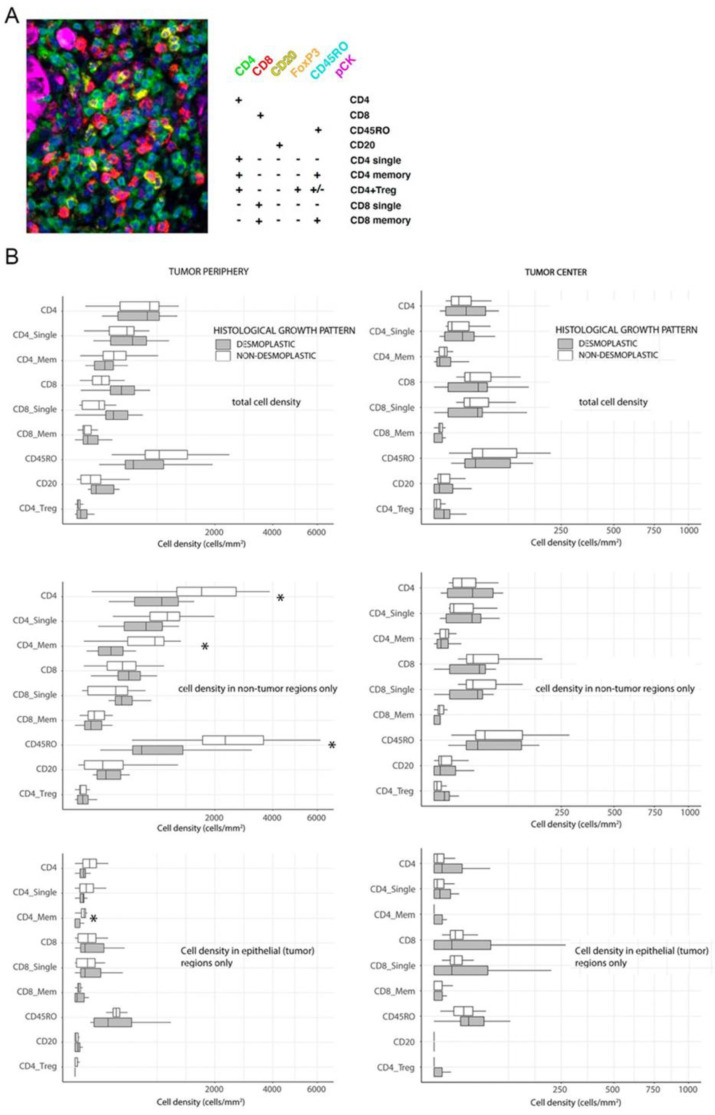
(**A**) Marker combination to identify each of the lymphocyte subsets assessed in the study. (**B**) Boxplots of cell densities (cells/mm^2^) for each of the subsets assessed and stratified by HGP. We differentiated cell counts according to different topographic locations in the tumor, either in the tumor periphery or in central tumor areas as well as differentiating whether infiltrates were located in the stroma or in tumor glands. Tumor periphery was defined as the tissue fraction from 100 μm of the proximal liver parenchyma until the first row of tumor cells, thus, including the capsule in dHGP CRC-LM and any rare portion of peritumoral stroma in ndHGP CRC-LM. Asterisks illustrate *p* value < 0.05, U-Mann Whitney test comparing dHGP vs. ndHGP.

**Figure 2 cancers-14-00689-f002:**
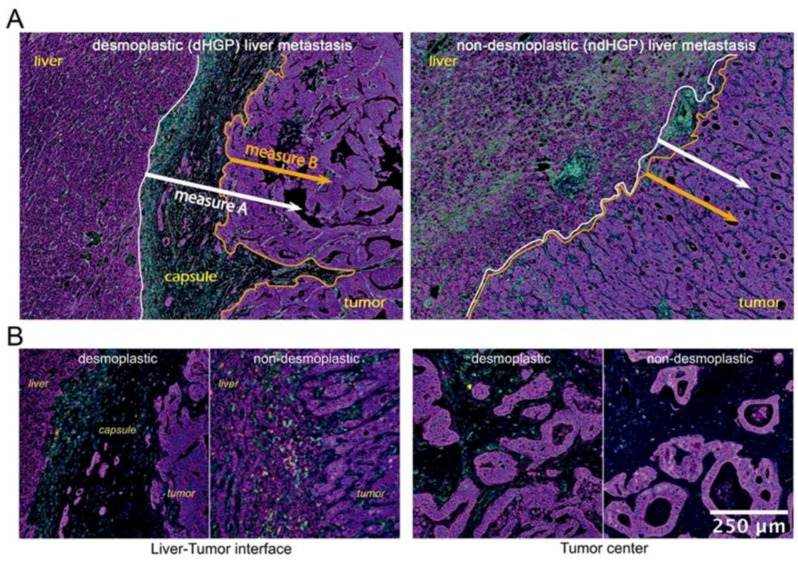
(**A**) Distances considered in relation to the tumor-host interface. Measure A, using as a reference the tumor-liver interface, including as part of the tumor, the fibrous capsule in case of _d_HGP or any small stromal area in the case of _nd_HGP; Measure B, using the tumor margin as a reference and then, not considering capsule or any peritumoral stromal. We use this measure throughout the manuscript to refer to the invasive margin. (**B**) Detailed magnification of the tumor/liver interface and central tumoral areas for both main histologic growth patterns.

**Figure 3 cancers-14-00689-f003:**
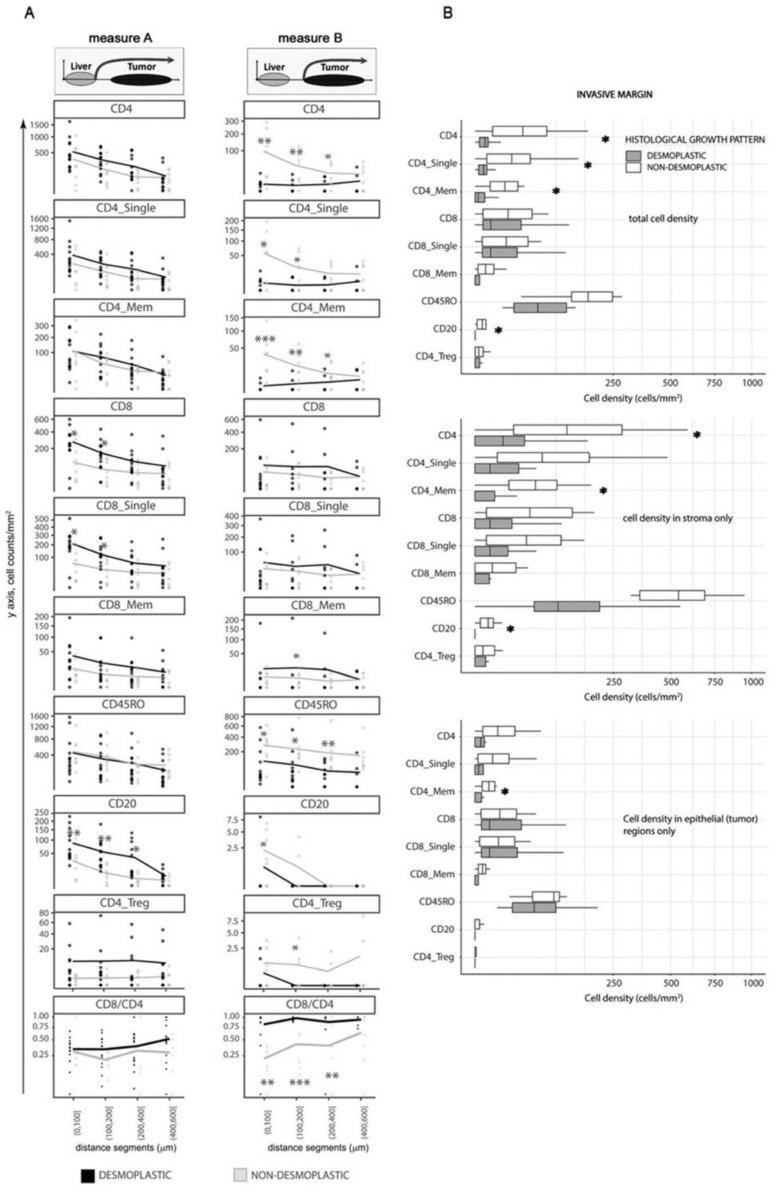
(**A**) Spatial assessment of lymphocyte subsets in _d_HGP and _nd_HGP CRC-LM according to two different measures, Measure A, from liver margin, Measure B, from the outer tumoral region (invasive margin). Lines show mean levels of cell densities. Dots show cell count/mm^2^. *X*-axis is square-root transformed (distant segments, μm). Mann–Whitney U test was applied for statistical analysis. (**B**) Boxplots of cell densities (cells/mm^2^) for each of the subsets were assessed and stratified by HGP according to the invasive margin. Asterisks illustrate *p* value < 0.05, Mann–Whitney U test comparing _d_HGP vs. _nd_HGP.

**Figure 4 cancers-14-00689-f004:**
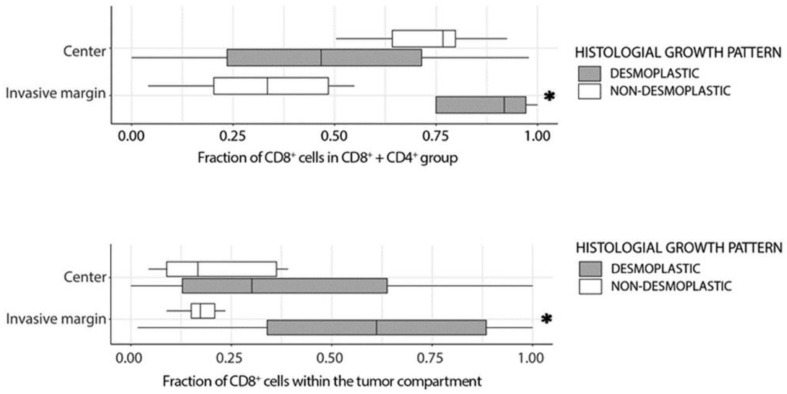
Top panel, boxplot for CD8/CD4 ratio comparing _d_HGP and _nd_HGP in central tumoral areas and at the invasive margin. Bottom panel, boxplot displaying the fraction of CD8^+^ cells within the epithelial compartment, both at the invasive margin and central tumor areas. Asterisks illustrate *p* value < 0.05, Mann–Whitney U test comparing _d_HGP vs. _nd_HGP.

## Data Availability

The data presented in this study are available in this article (and Appendix A).

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
