# Peer review of "Spatial Immunology in Liver Metastases from Colorectal Carcinoma according to the Histologic Growth Pattern"

_cancers, 2022, doi:10.3390/cancers14030689_

Round 1

Reviewer 1 Report

The manuscript by Garcia-Vicién and colleagues examines the lymphocytic infiltrates of liver metastases derived from colorectal cancer (CRC). Specifically, the authors separated desmoplastic (fibrotic capsule surrounding the metastases) and non-desmoplastic metastases and spatially characterised the distribution of lymphocytes in the marginal and central areas of the metastases. The authors show that both types of metastases have similar density of CD8+ cells in the intratumoral regions. However, in encapsulated metastases CD8+ T cells are in closer contact with tumour cells while they remain in the stromal areas in non-encapsulated metastases.

This is an interesting and well-conducted study, which may be of interest to a wide readership. However, the authors could implement some automated algorithms that have been published for “histocytometry” approaches.

I also have some additional comments that need to be addressed:

1) Why do the authors start with figure 2 in the text? Also figure 1A is not mentioned in the results section.

2) Figure 2: Representative multicolour /composite pictures should be shown of ndHGP and dHGP metastases with additional magnification of the margins and centers of both metastasis types.

3) The authors use a morphological/histological approach to perform spatial analysis of lymphocytic infiltrates. However, many recent studies use a more automated approach to distance analysis and neighbourhood analysis. I personally like the approach of performing these analyses with CytoMAP (https://doi.org/10.1016/j.celrep.2020.107523). Microscope pictures can be imported into FlowJo and resulting .csv files can be imported into CytoMAP.

4) A few comments to the text:

-Simple summary (lines 19-28): HGP is introduced for the first time, but the full name is not provided. Some spaces are also missing after the sentences.

- Line 62: Bracket is not closed after dHGP

- Line 109-110: What does “we need illustration of ‘old’ selections” mean?

-Lines 153 and 163: CD4_Treg. Do the authors mean “CD4+ Tregs” or “CD4- Tregs”?

-Line 165: Change “CD20 B cells” to “CD20+ B cells”

-Line 184: Please start sentence with a capital letter

-Line 213: Please include reference instead of “(reference)”.

-Line 322: What does “consecutive cases” mean in this context?

5) Please explain formula of CD8/CD4 ratio better. Why is CD8 on both sides of the formula?

6) Figure 3A: Please add labels of the x/y axes.

7) Figure 4: Please change the label “fraction of CD8+ cells in CD8+CD4+ group, as it implies that there are cells that are double positive for CD8 and CD4.

8) Macrophages are not included in the analysis, but it would definitely be worth to study them in this context. Macrophages are heavily involved in cancer-related inflammation of solid tumours. In addition, liver metastases recruit immunosuppressive macrophages that promote antigen-specific T cell apoptosis within the liver. This would be particularly interesting to study in the ndHGPs where the authors see an immunosuppressive microenvironment that blocks the cytotoxic activity of CD8+ cells.

Author Response

We uploaded a point-by-point response to the previous reviewer's comments.

Reviewer 2 Report

The authors present on lymphocytic infiltrates in22  colorectal cancer liver metastases (CRC-LM) patients, using multiplex immunohistochemistry and spatial analysis.

They found a difference in the histological growth patterns (HGP) for CD8+ cells, but with a p value of 0.053, that means it was not significant, NOT 'marginal statistical significance'.

The authors report they found significant differences for CD4+, CD4+ memory, CD45RO in stroma and CD4+ memory in epthelial, but do not mention the p values in the text, but rather state p<0.05 in Figure 2.

Is there any practical application for these findings in the authors view for the future?

I think the wording could be far clearer and to the point.

Author Response

We uploaded a point-by-point response to the previous reviewer's comments

Round 2

Reviewer 1 Report

The manuscript by Garcia-Vicién and colleagues examines the lymphocytic infiltrates of liver metastases derived from colorectal cancer (CRC). This is an interesting and well-conducted study, which may be of interest to a wide readership.

The authors addressed all my comments during the revision or answered my questions convincingly.

I have an additional note on my original review comment 6: Figure 3A: Please add labels of the x/y axes.

It is not necessary to add a label to each individual plot. Since you oriented the plot in a way that all y and x axes are aligned, I would add a representative label for the y-axis on the side (centered). For the x-axis I would add the label at the bottom 2 panels.

Author Response

As suggested by  the reviewer we modified the graph including the Y and X axis.

Reviewer 2 Report

The authors have made some efforts to improve the manuscript.

However, p>0.05 is not significant. and any other interpretation is wrong. Otherwise, why do statistics?

There is a lot of interesting information in this paper.

Still requires extensive re-editing.

Author Response

Regarding the concern about the p value, we removed completely the sentence. We hope the change made satisfy the reviewer. We agree that p > 0.05 is not significant even when the value is very close to 0.05.

In our opinion, what we wanted to capture in the document is, in each section of the results, to introduce the question/problem, address it, and summarize the conclusions for each of the sections. We think it makes reading easier and helps us to better explain the sequence of objectives set and their resolution.